# Effect of Different Additives in Diets on Secondary Structure, Thermal and Mechanical Properties of Silkworm Silk

**DOI:** 10.3390/ma12010014

**Published:** 2018-12-20

**Authors:** Lan Cheng, Huiming Huang, Jingyou Zeng, Zulan Liu, Xiaoling Tong, Zhi Li, Hongping Zhao, Fangyin Dai

**Affiliations:** 1State Key Laboratory of Silkworm Genome Biology, Key Laboratory of Sericultural Biology and Genetic Breeding, Ministry of Agriculture, College of Biotechnology, Southwest University, Chongqing 400715, China; chenglan2018@swu.edu.cn (L.C.); xltong@swu.edu.cn (X.T.); 2Institute of Biomechanics and Medical Engineering, Tsinghua University, Beijing 100084, China; hhm207@163.com; 3College of Biology, Hunan University, Changsha 410082, China; jingyouz@hnu.edu.cn; 4Chongqing Engineering Research Center of Biomaterial Fiber and Modern Textile, College of Textile and Garment, Southwest University, Chongqing 400715, China; 13658349461@163.com (Z.L.); tclizhi@swu.edu.cn (Z.L.)

**Keywords:** silkworm silk, additives, secondary structure, thermal and mechanical properties

## Abstract

In this study, eight types of materials including nanoparticles (Cu and CaCO_3_), metallic ions (Ca^2+^ and Cu^2+^), and amino acid substances (serine, tyrosine, sericin amino acid, and fibroin amino acid) were used as additives in silkworm diets to obtain *in-situ* modified silk fiber composites. The results indicate that tyrosine and fibroin amino acids significantly increase potassium content in silk fibers and induce the transformation of α-helices and random coils to *β*-sheet structures, resulting in higher crystallinities and better mechanical properties. However, the other additives-modified silk fibers show a decrease in *β*-sheet contents and a slight increase or even decrease in tensile strengths. This finding provides a green and effective approach to produce mechanically enhanced silk fibers with high crystallinity on a large scale. Moreover, the modification mechanisms of these additives were discussed in this study, which could offer new insights into the design and regulation of modified fibers or composites with desirable properties and functions.

## 1. Introduction

Silk fibers from *Bombyx mori* silkworms are receiving significant attention for various functional textiles, biomaterials, and nanodevices, due to their outstanding mechanical properties, biocompatibility, and degradable characteristics [1,2,3,4,5]. Currently, tremendous efforts have been made to fabricate enhanced or functionalized silk fibers or silkworm cocoons through various physical [6,7,8,9,10,11,12,13,14], chemical [15,16,17,18], and biological processes [19,20,21,22]. It is of particular interest that direct feeding silkworm larvae with a series of additives (e.g., organic dyes [23,24,25], nanoparticles (NPs) [9,10,26,27,28,29], unnatural amino acids [30], as well as bordeaux mixture [31]) is a green and sustainable *in-situ* functionalization approach to modify silk fibers. In this case, these feeding additives mainly distribute in silk fibroin rather than silk sericin, which can improve the thermal and mechanical properties of silk fibers [10,24,32]. Nevertheless, research [9,10,28] has indicated that these additives hinder the conformational transition from random coil/*α*-helix to *β*-sheet of silk fibroin and reduce the degree of crystallinity. This implies that the secondary structures and mechanical properties of silk fibers could be further optimized and improved through the feeding of appropriate additives.

The secondary structures and mechanical properties of silk fibers have been shown to be influenced not only by their unique amino acid sequences [33,34] and contents [35,36], but also by the conformational transition of silk protein and spinning condition [3,37,38]. The natural spinning process of silk involves a conformational transition in silk protein from the random coil and/or *α*-helix conformations into *β*-sheets under shear or rapid elongation flow, which can be influenced by pH and metallic ions, as well as shear force and spinning speed [37,38,39,40,41,42,43]. Previous *in vitro* and *in vivo* study [38] revealed that these important metallic ions, including Ca^2+^, Cu^2+^, K^+^, and Zn^2+^, can induce the conformational transition of silk fibroin towards the *β*-sheet formation. Meanwhile, research [21,44] has indicated that silk fibers with superior strength and extension capacity can be obtained through disrupting the ion balance in the spinning duct. Different types of additives, such as nanoparticles, ionic compounds, and amino acid substances, may have different influences on the ion balance of Ca^2+^, Cu^2+^, K^+^, and Zn^2+^ in the spinning duct, leading to significant differences in the thermal and mechanical performance of silk fibers.

However, it is currently unclear how different types of additives regulate these metallic ions and the secondary structures in silk fibers. Most of the recent studies focused on the modification of silk fibers by feeding nanoparticles (NPs). Few studies on the modification of silk fibers by feeding ionic compounds or organic amino have been conducted. Thus, the present study investigates the influence of different types of additives, including nanoparticles, ionic compounds, and organic amino, in silkworm larvae diets on the conformational transition, thermal, and mechanical property of silk fibers. For nanoparticles, the two typical NPs (Cu and CaCO_3_) are used; for ionic compounds, two corresponding ionic compounds (CaCl_2_ and CuCl_2_) are used to supply Ca^2+^, Cu^2+^; for organic amino, serine (Ser), tyrosine (Tyr), sericin amino acid (SAA), and fibroin amino acid (FAA) are selected as the representative research material. The eight materials were used as additives to feed silkworm larvae at the fifth instar to fabricate modified silk fibers. The effects of these additives on the contents of Ca^2+^, Cu^2+^, Zn^2+^, and K^+^ in silk fibers, which play a key role in the conformational transition of silk fibroin towards *β*-sheet formation, were first examined by inductively coupled plasma mass spectroscopy (ICP-MS). The secondary structure of silk fibers was then characterized by Fourier-transform infrared spectroscopy (FTIR) and X-ray diffraction (XRD). Furthermore, thermogravimetric analysis (TGA), quasistatic and continuous dynamic tensile tests were conducted to evaluate the thermal and mechanical properties of modified silk fibers. The relationships among the metallic ions, conformational transition, and the thermal and mechanical property of the resulting silk fibers in this study provide important insights into the design and development of reinforced silk fibers.

## 2. Materials and Methods

### 2.1. Materials

Eight different materials (shown in Table 1), including NPs, metallic ions and amino acid materials were used as additives to directly feed silkworm larvae. Approximately 0.2 g of each additive was mixed with 100 mL of water and then sprayed onto fresh mulberry leaves to thereby constitute modified diets. Silkworms (Chinese strain 932) were obtained from State Key Laboratory of Silkworm Genome Biology (Southwest University, Chongqing, China) and separated into several groups (30 samples for each group), and then fed with modified diets from the first day of their fifth instar stage until complete cocooning. The obtained silkworm cocoons from normal and modified feedings were reeled with the normal reeling process to collect silk fibers [10]. The sectionalized silk fibers were then degummed twice in 0.5% Na_2_CO_3_ solution for 45 min and silk fibroin fibers were obtained after rinsing with deionized water and dried at 50 °C in an oven. It should be mentioned that, for the following tests for each group, we chose silk fibers from five cocoons with almost the same lengths and weights in nearly the same position (from the 200th to 600th meter) as samples to reduce the variability at the intraspecific and intra individual levels. All the materials used in the following experiments are degummed silk fibers.

### 2.2. ICP-MS Measurements

Silk fiber samples were dried at 60 °C for 12 h and digested with 10 mL of 14.4 mol/L HNO_3_, then the different metal element concentrations in the fibers were determined using an inductively-coupled plasma mass spectrometer (ICP-MS, 7500ce, Agilent Technologies, Santa Clara, CA, USA) at an atomizing chamber temperature of 2 °C and a radio frequency power of 1500 W. All measurements were repeated three times and then averaged.

### 2.3. FTIR Analysis

FTIR spectra of the obtained silk fibers were analyzed using a VERTEX 70v spectrometer (Bruker Corporation, Karlsruhe, Germany) using KBr pellets. All spectra were ascertained using a reflection mode, with a resolution of 4 cm^−1^ over the range of 1000 cm^−1^ to 1800 cm^−1^. The deconvolutions of the Amide I region (1600 cm^−1^ to 1720 cm^−1^) were performed using the Peakfit 4.0 software (Peakfit 4.0, Systat Software, Middlesex, UK) for quantitative analysis of the secondary structure of the silk fibers. The intensity of the amide I band was corrected by subtracting a straight baseline at the bottom of the peak, and the peak was fitted with a Gaussian function. The deconvoluted Amide I spectra was area-normalized, and the relative contributions of the crystalline (*β*-sheet) bands were used to determine the degree of crystallinity [9,10,45,46].

### 2.4. XRD Analysis

Wide-angle XRD measurements were performed on a D8-Advance XRD diffractometer (Bruker, Karlsruhe, Germany) at a scan rate of 4° 2*θ* min^−1^ from 10° to 40° with Cu Kα radiation. The irradiation conditions of the X-ray generator system were 40 kV and 30 mA. The crystallinity of silk fiber was calculated from the area ratio between the crystalline and amorphous regions of 2*θ* diffractograms using the MDI Jade 6.0 software (Jade 6.0, MDI, Livermore, CA, USA).

### 2.5. TGA Analysis

Thermal properties of the silk fibers were evaluated using a Q5000 TG analyzer (TA Instruments, New Castle, DA, USA) from ambient temperature to 80 °C with a heating rate of 10 °C/min. Nitrogen was used both as a balance gas and sample gas at a flow rate of 50 mL/min. Derivative thermogravimetric (DTG) curves were plotted from the data extracted from the TGA.

### 2.6. Mechanical Properties of the Silk Fibers

Quasistatic and continuous dynamic tensile tests of degummed silk fibers were conducted using an Agilent T150 UTM Nano tensile tester (Agilent Technologies, Santa Clara, CA, USA), using ambient conditions. Continuous dynamic analysis for each sample was conducted using a dynamic force amplitude of 4.5 mN and at a frequency of 20 Hz. Prior to testing, all the samples were mounted under tension into 6-mm gauge length cardboard frames, and the average diameter of the fibers was measured at 10 different positions using a laser scanning microscope (VK-X100K, Keyence, Osaka, Japan). The experimental stress-strain curves were then recorded during quasistatic tensile tests to obtain Young’s modulus *E*, elongation *ε*, ultimate tensile strength *σ*_b_, and strain energy density *w*. In addition, dynamic storage modulus *E*′, loss modulus *E*′′, and loss factor tan *δ* were also assessed using the continuous dynamic tensile tests. Thirty individual samples from each group were tested.

### 2.7. Statistical Data Analysis

All statistical data were expressed as the mean ± standard deviation. Statistical analyses were conducted using one-way ANOVA as implemented by the SPSS statistical software. A *p*-value of <0.05 compared with the control group was considered statistically significant.

## 3. Results and Discussion

### 3.1. Contents of Metal Elements in Silk Fibers

Metallic ions, including K^+^, Ca^2+^, Cu^2+^, and Zn^2+^, are the requisite substances in the silkworm body, and they can affect the conformation of silkworm silk fibers *in vitro* and *in vivo* [38,40,47,48,49]. Thus, the distributions of these four metal elements in silk fibers as modified by different additives were quantitatively investigated by ICP-MS (Figure 1). The results indicate that metal element levels vary among different modified silk fibers. Potassium and calcium are the most abundant elements in silk fibers, reaching above 100 mg/kg, whereas copper and zinc are below 20 mg/kg, which is in agreement with the findings of previous studies [44,50]. The copper element-contained or calcium element-contained additives are absorbed by silkworm larvae through their diets and eventually spun into silk fibers. The average copper and calcium levels in the control silk fibers are 3.80 mg/kg and 1540 mg/kg, respectively, whereas copper contents increase to 8.38 mg/kg and 8.50 mg/kg in the silk fibers modified by Cu NPs and CuCl_2_, and calcium contents to 1787 mg/kg and 1920 mg/kg in the two silk fibers modified by CaCO_3_ NPs and CaCl_2_, respectively. Moreover, metallic ions are easier to incorporate into silk fibers than NPs, as there are more corresponding metal elements in metal ion-modified silk fibers. In addition, silk fibers modified by Cu NPs, Cu^2+^, and Ser show a higher calcium content, whereas those modified by Tyr, SAA, and FAA exhibit lower calcium levels compared to the control silk fibers. The additives without copper components reduce the copper content of silk fibers, particularly for the modified fibers by Ca^2+^ additives, and only 0.96 mg/kg copper in silk fibers is detected by ICP-MS. In terms of zinc element content in silk fibers (Figure 1c), the Cu NPs-modified silk fibers show the highest levels, in which zinc content is 7.84 mg/kg, followed by the Cu^2+^-modified silk fibers (6.01 mg/kg), FAA (5.60 mg/kg), Ser (5.30 mg/kg), SAA (4.60 mg/kg), and CaCO_3_ NPs (4.48 mg/kg), all of which have higher levels than the control silk fibers (3.74 mg/kg). However, the two kinds of silk fibers modified by Ca^2+^ and Tyr show a lower zinc content. More importantly, amino acid materials significantly increase the potassium content of silk fibers. Figure 1d illustrates that the modified silk fibers by Ser, Tyr, SAA, and FAA fed additives have more than 3000 mg/kg potassium, whereas that of the control silk fibers is only about 365 mg/kg. Nevertheless, the observed changes in the potassium content in NPs- and metallic ion-modified silk fibers are not highly significant.

The results of ICP-MS demonstrate that Cu-containing additives, either as NPs or ions, increase the copper and zinc contents of silk fibers. This specific increase in copper or zinc content may infer the concentration enhancement of the corresponding metal ion in the silk glands to coordinate with silk fibroin macromolecules, leading to further *β*-sheet aggregation of functional proteins [49]. Meanwhile, an increase in calcium content in silk fibers also proves the existence of CaCO_3_ NPs and Ca^2+^. The contents of three other metal ions in Ca NPs- and Ca^2+^-modified silk fibers are relatively lower than those in the control silk fibers, indicating that the additives inhibit *β*-sheet transition [50]. In silk fibers modified by amino acid materials, a 10-fold increase in potassium content and a significant decrease in calcium content are observed compared to that in the control silk fibers. These variations may help in the breakdown of the gel network and allow silk fiber macromolecular chains to move more freely, thereby enabling the protein dope to flow through the anterior division, thus promoting subsequent *β*-sheet formation [38]. The different variations in the four kinds of metal ions in silk fibers caused by the different additives may relate to biological, chemical or physical processes in the silkworm body. Further study should be systematically conducted for a deep understanding of the bio-interaction between silkworm larva and additives.

### 3.2. Secondary Structure of Silk Fibers

FTIR was used to study the effects of additives on the secondary structure of silk protein, as shown in Figure 2a. In the band region from 1800 to 1000 cm^−1^, there are three typical amino bands including amide I (1720–1600 cm^−1^), amide II (1600–1500 cm^−1^), and amide III (1350–1200 cm^−1^) [45,46,51]. It is obvious that the additives in silkworm feeding do not significantly change the basic secondary structure fibroin protein from the similar identical peak positions in the band region. For better understanding the influences of the additives on the crystalline structure, the Amide I region (the small figure in the Figure 2a) is used for detailed analysis. The identical peak at 1640 cm^−1^ is attributed to the *β*-sheet configuration, whereas the peaks at approximately 1665 cm^−1^ and 1706 cm^−1^ are assigned to random coils and *β*-turns, respectively. In terms of the spectra of Tyr- and FAA-modified silk fibers, the Amide I band shows very strong peaks at around 1636 cm^−1^, whereas the other silk fibers have relatively weak peaks at around 1640 cm^−1^. These two additives may have induced changes in the silk proteins from a random to a crystal configuration. Furthermore, to better understand the influence of dietary additives on silk protein properties, Fourier self-deconvolution (FSD) of amide I region is conducted to quantify the contents of random coil, *β*-sheet, and *β*-turn to determine the degree of crystallinity (Figure 2b). The silk fibers from nine groups do not exhibit distinct differences in terms of random coil content; however, significant variations are observed in both *β*-sheet and *β*-turn content, particularly for the Tyr- and FAA-modified groups. The *β*-sheet contents of Tyr- and FAA-modified silk fibers are about 48% and 53%, respectively, which are markedly higher than that of the control group (35%). *β*-turn content, in contrast, is 13% for the control silk fibers, and around 4% for both Tyr- and FAA-modified silk fibers.

XRD was used to confirm the conformational transition and crystallinity of these silk fibers. Diffractograms of nine types of silk fiber samples are presented in Figure 2c. All samples have similar strong peaks at around 2*θ* = 21°, 24°, 27°, and 35°, representing the characteristic of the *β*-sheet crystals of silk fibroin [28,51,52], thereby proving that the silk fibers with or without additives have similar secondary protein structure. Simultaneously, the crystallinity values of each sample obtained from the Jade 6.0 software (Jade 6.0, MDI, Livermore, CA, USA) are summarized in Figure 2d. The crystallinity of the nine types of silk fibers varied from 30–49%. Compared with the control silk fiber, with an overall crystallinity of 36%, increases in crystallinity can be found in silk fibers modified by Tyr (39.8%) and FAA (48.5%), which coincide with the results of FTIR analysis. However, the overall crystallinity values in the other six modified fibers are relatively lower than that of the control fibers.

Physical properties, particularly the strength and stiffness of silk fibers, are mainly determined by *β*-sheet crystallite, including the crystal sizes, aspect ratio, and distribution [3,37]. The FTIR and XRD results suggest that the presence of NPs or metallic additives in silk protein hinders the transformations of *α*-helices and random coils to *β*-sheet structures due to the decrease of *β*-sheet crystallite. Other additives such as TiO_2_ [28], single-walled carbon nanotubes (SWNT), and grapheme (GR) [9] also induce a slight decrease in the *β*-sheet contents in modified silk fibers. It may be attributable to steric hindrance effects, metallic environment effects, or noncovalent interactions between materials and silk fibroin through physisorption or chemisorption [9]. In particular, *β*-sheet crystallite content increases in the silk fibers obtained from Tyr- and FAA-modified diets, particularly changes in the amino acid sequence GAGAGS that comprise crystalline regions [34,36,53].

### 3.3. Thermal Stabilities of Silk Fibers

The silk fibers from the nine groups were subjected to TG analyses from ambient temperature to 800 °C in the nitrogen atmosphere (Figure 3a,b). Furthermore, to better understand the effects of different additives on the thermal stability of silk fibers, the temperature at which the fibers start losing weight and the temperature at which a maximum weight loss rate occurred, as well as residue mass, are individually compared (Table 2). The characteristics of thermal degradation in all types of silk fibers are almost the same, which implies that the additives in the silkworm diet do not induce significant changes in the basic thermal degradation process. In all samples, initial weight loss is observed at approximately 100 °C, which corresponds to the temperature at which water evaporates. The results show that silk fibers modified by fed additive Ser and SAA exhibited higher thermal stability compared to control silk fibers. In these two samples, abrupt decreases occur beyond 294 °C and 297 °C, respectively, producing two degradation peaks at around 324 °C. These losses occur at lower corresponding temperatures for the control silk fibers at 293 °C and 323 °C, respectively. In addition, approximately 21% and 19% of the initial weights of Ser group and SAA group remained at the end of the run at 800 °C, whereas the residue mass of control silk fibers is only 12%. However, significant decreases in mass are observed in the CaCO_3_ NPs-and Tyr-modified silk fibers in the range of 600 °C–800 °C, and the residue mass is around 3%. The decrease of the residue mass may result from the changes of bond energy or content of amino acid in silk fibers, which could be further studied systemically.

### 3.4. Mechanical Properties of Silk Fibers

The static and dynamic mechanical properties of the silk fibers were evaluated using quasistatic and continuous dynamic tensile tests. Representative stress-strain curves of each sample from quasistatic tensile tests were obtained and are shown in Figure 4a. Static mechanical properties were then calculated from stress-strain curves and summarized in Figure 4b,c. The ultimate tensile strength of the modified silk fibers increases to varying degrees compared to the control silk fibers (332 MPa), except for the fibers obtained by Ca^2+^ feeding. Specifically, the corresponding mean values of the silk fibers obtained by fed additive Cu NPs, Ser, Tyr, and FAA are 378 MPa, 380 MPa, 415 MPa, and 412 MPa, respectively. Moreover, FAA-modified silk fibers exhibit a significantly higher elongation value of 21%, whereas those of the other samples are within the range of 15–19%. As for the index of Young’s modulus, silk fibers from the SAA diet exhibit the highest value (12.6 GPa). However, obvious decreases in Young’s modulus are observed in metallic ion-modified silk fibers. As a consequence of the differences in Young’s modulus, elongation, ultimate tensile strength, and the calculated strain energy density vary among different additives. The value of strain energy density is 63 MJ/m^3^ for FAA-modified silk fibers and 59 MJ/m^3^ for the Tyr-modified group, which is significantly larger than that of the control group (42 MJ/m^3^). The improved static mechanical properties of modified silk fibers obtained from FAA or Tyr additives are highly related to the increase of *β*-sheet contents and crystallinity.

Traditionally, quasistatic tensile tests measure how much energy is absorbed by fibers when these are extended, and the function of how fibers absorb energy can be achieved from continues dynamic analysis (CDA) [54,55,56,57]. A slight oscillating strain is loaded on silk fiber when it is extended, and dynamic stress response to the oscillating strain can be measured. The two typical indices corresponding to the elastic and viscous behavior of fibers, namely, storage modulus (*E*′) and loss modulus (*E*′′), can provide quantitative information on how variations in the structure of polymer chains affect the mechanical properties of fibers. The two parameters can be calculated at every point during the tensile test. Loss tangent (tan*δ*), which is the ratio of loss modulus (*E*′′) to storage modulus (*E*′), will vary with alterations in how fibers store energy elastically, how fibers dissipate energy as heat, or both. CDA investigations were performed to further understand the influence of additives on the molecular structure of silk fibers, as well as storage and loss moduli as a function of tensile strain. Typical storage modulus-strain curves and loss modulus-strain curves were presented in Figure 5a,b, respectively. Mechanical indices, such as dynamic ultimate tensile strength and storage and loss moduli, are also assessed (Table 3). The dynamic tensile behaviors of all tested silk fibers showed similar curves for the storage and loss moduli and changed as the fibers are strained, which is similar to the behavior of spider silk [54]. During the first 0.8–2% strain, an initial drop in storage modulus is observed, which is predicted by the breaking of the H-bonds during fiber extension within the elastic region, and a rapid increase in loss modulus, which coincide with the frictional heat generated as the mobility of the amorphous chains increases during yield. After the yield point, the storage modulus increased as amorphous chains begin to align along the axis of extension while the loss modulus increases due to frictional forces. The yield point of silk fibers obtained from the FAA diet is about 2%, which is higher than that of other silk fibers, implying that the hydrogen bonding in silk fibers is relatively hard to break [54,56]. Table 3 shows that the modified silk fibers can be classified into two groups, compared with control silk. One group exhibits significantly higher strength and modulus, including Cu NPs, Ser, Tyr, SAA, and FAA, with an overall mean range of 346–459 MPa for tensile strength, 13.1–15.7 GPa for dynamic modulus, 18–20 GPa for storage modulus, and 1.48–1.96 GPa for loss modulus. The other group containing Cu^2+^, CaCO_3_ NPs and Ca^2+^-modified silk fibers display inferior values, ranging on average from 280–333 MPa for tensile strength, 11.3–12.7 GPa for dynamic modulus, 15.1–15.8 GPa for storage modulus, and 1.32–1.45 GPa for loss modulus. Most importantly, silk fibers modified by SAA and FAA present relatively higher dynamic tensile strength compared to quasi-static one, implying a more stable molecular structure for silk fibers. Changes in loss tangent in silk fibers are not evident, ranging from 0.09–0.10, similar to that of spider silk fibers [54]. All these materials indicate an immense capacity to dissipate kinetic energy, indicating their potential for application as reinforcement in composite and impact resistant materials. Combining the results of metal ions contents in the modified silk fibers, secondary structure, and the thermal and mechanical properties of the modified silk fiber, it is revealed that Tyr or FAA as additives to modify silk fibers can significantly increase potassium content in silk fibers and induce the transformation of α-helices and random coils to *β*-sheet structures, resulting in higher crystallinities and better mechanical properties. However, the other additives-modified silk fibers show a decrease in *β*-sheet contents and slight increase or even decrease in tensile strengths.

The secondary structures and properties of modified silk fibers vary with different additives. For example, both organic dyes and NPs hindered the formation of *β*-sheets as well as crystallites, despite the observed improvements in the mechanical properties of NPs-modified silk fibers [9,10,28,58]. The detailed relationship between the ratios of *β*-sheet content and mechanical properties (toughness as a represent) of silk fibers between the modified and control groups of previous studies and the present investigation are summarized in Figure 6. Most additive-modified silk fibers exhibited a toughness ratio above one, with a *β*-sheet content ratio below one. These additives with the studied concentrations can improve the mechanical properties of silk fibers but hinder *β*-sheet conformational transitions. The incorporation of excessive materials such as TiO_2_ NPs, SWNT, or GR also resulted in the deterioration of the mechanical properties of silk fibers, as well as low *β*-sheet content. In particular, Tyr and FAA used in this study not only led to an increase in toughness but also induced the conformational transition of silk fibroin from random coils to *β*-sheet. This is the first report that describes the positive effects of additives on both *β*-sheet conformation transition and mechanical properties. Therefore, further efforts should be made on elucidating the molecular mechanism of Tyr and FAA feeding for silkworm larvae in relation to low biotoxicity and cost in producing high-crystalline and strong silk fibers.

This systematic study on the effects of NPs, metallic ions, and amino acid substances on the secondary structure and physical properties of silk fibers provide us with a better understanding of the relationship between additives and silk fibers. The effects of various additives on the secondary structure of silk fibers are illustrated in Figure 7 and *β*-sheet content is used to present the secondary structure of silk fibril. The possible modification mechanism varies from NPs, metallic ions, and amino acid substances. The fed NPs act as reinforcements to increased mechanical properties in the enhanced silk fiber, even though there was an inhibition effect on the *β*-sheet conformational transitions. Metal ions such as Ca^2+^ and Cu^2+^ changed the mechanical properties of silk fibers by disrupting the ion balance in silk protein. Besides, excessive metal ions, especially Ca^2+^ in the silk gland, are unfavorable for silk fibroin conformation. Nevertheless, some amino acid additives improve the mechanical properties of the modified silk fibers by increasing *β*-sheet content, which may involve the synthesis and secretion of proteins. The three possible effects mechanism of additives on the structure of silk fiber provide us with guidance for designing the mechanical properties of silk fibers. There are still several important and interesting questions that cannot be answered by our current work, such as the reason for the change in various metal ions in fiber due to change in diet, the stability of additives incorporated in fibers when soaked in solutions, and the detailed biological process. Further studies regarding these interdisciplinary questions are thus warranted and meaningful.

## 4. Conclusions

In summary, eight types of materials, including NPs, metallic ions, and organic amino acid were used as additives in silkworm diets to produce *in-situ* modified silk fibers. The impacts of these additives on metallic element content, conformational transition, and the thermal and mechanical property of silk fibers were systematically investigated to reveal the reinforcement mechanisms of different additive materials. Our findings indicate that tyrosine and fibroin amino acids significantly increase potassium content in silk fibers and induce the transformation of *α*-helices and random coils to *β*-sheet structures, resulting in higher crystallinities and better mechanical properties. However, the other additives-modified silk fibers show a decrease in *β*-sheet contents and a slight increase or even decrease in tensile strengths. The results of the present study suggest that the enhanced mechanical property of silk fibers modified by nanoparticles may be attributable to the high performance of inorganic particles themselves, whereas the amino acid-modified silk fibers exhibit a change in potassium content to increase silk crystallinity, and therefore induce their improved mechanical property. The findings indicate that specific amino acid, such as tyrosine and fibroin amino acids, are the best additives because they are not only cheap and safe but can also increase both the crystallinity and mechanical property of silk fibers. In addition to this, the mechanism effects of additives on the secondary structure and physical properties of silk fiber provide us with a guide for designing silk materials with favorable properties or functions.

## Figures and Tables

**Figure 1 materials-12-00014-f001:**
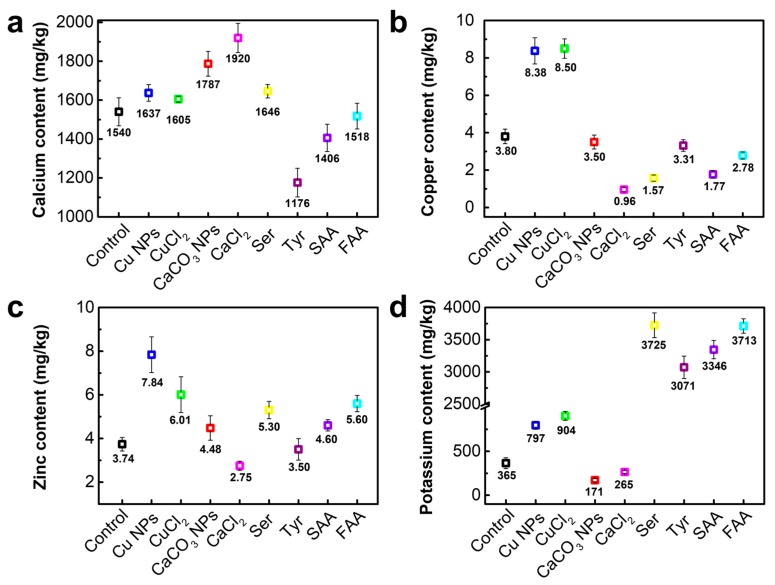
Distributions of four metallic elements in silk fibers. (**a**) Calcium content, (**b**) Copper content, (**c**) Zinc content, (**d**) Potassium content. The X axis is silk fiber samples obtained from different diet additives and the control is the normal silk obtained from the traditional diet without any additives. Data are mean ± sd. (Ser = serine, Tyr = tyrosine, SAA = sericin amino acid, FAA = fibroin amino acid).

**Figure 2 materials-12-00014-f002:**
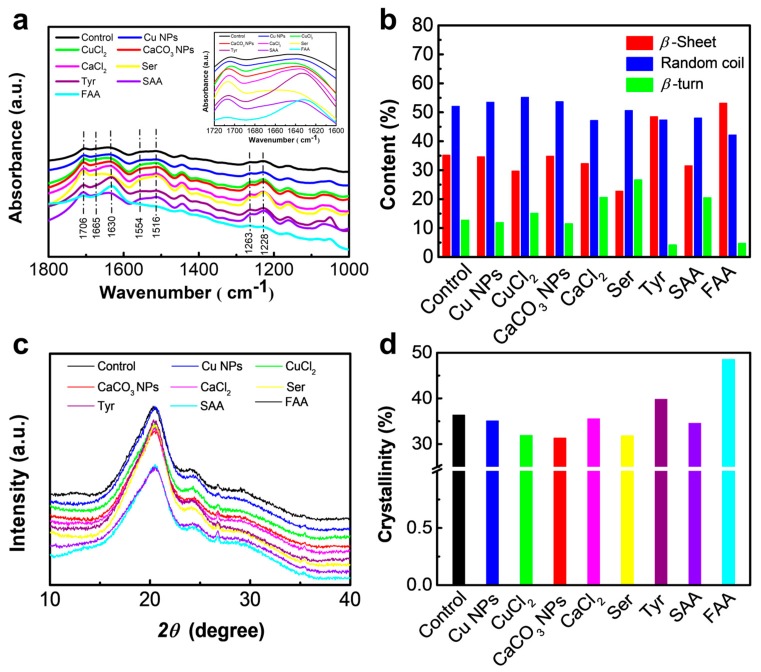
Secondary structure of silk fibers. (**a**) Fourier-transform infrared (FTIR) spectra of silk fibers obtained from normal and modified diets and the small image is the FTIR spectra in amide I band; (**b**) content of *β*-sheet, *β*-turn and random coil after deconvolution of FTIR spectra in amide I band. (**c**) X-ray diffraction (XRD) spectra of silk fibers; (**d**) Crystallinity percentage of silk fibers. The X axis for (**b**,**d**) is silk fiber samples obtained from different diet additives and the control is the normal silk obtained from traditional diet without any additives. (Ser = serine, Tyr = tyrosine, SAA = sericin amino acid, FAA = fibroin amino acid).

**Figure 3 materials-12-00014-f003:**
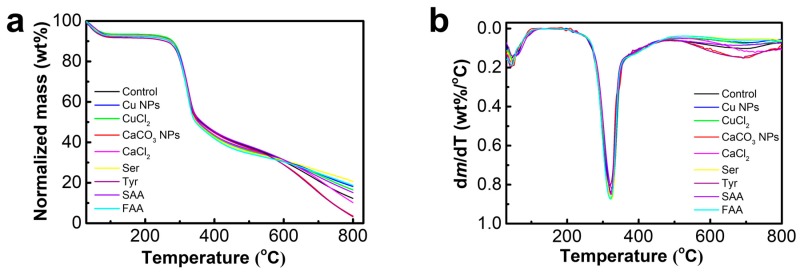
TGA results of silk fibers. (**a**) TG curves; (**b**) DTG curves.

**Figure 4 materials-12-00014-f004:**
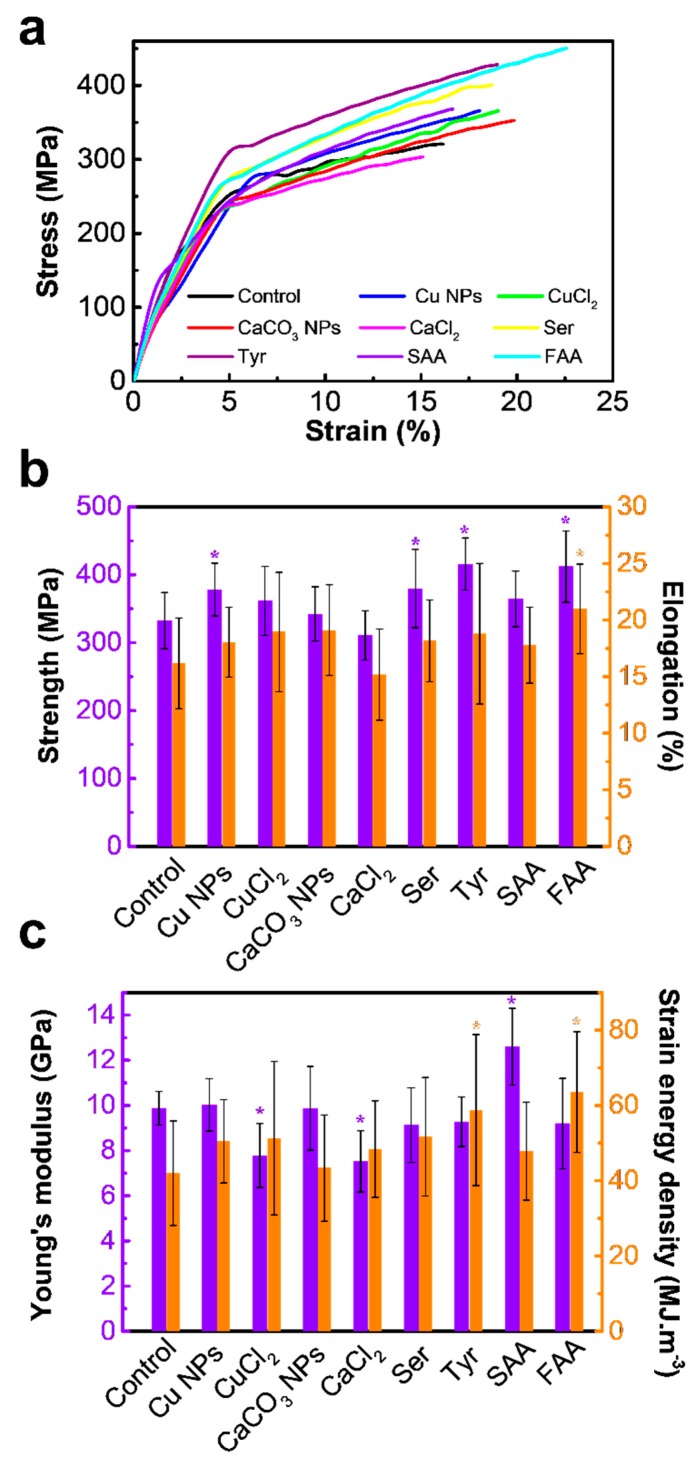
Quasi-static tensile behaviour of silk fibers. (**a**) Typical stress-strain curves, (**b**,**c**) are mechanical properties of tensile strength, elongation, Young’s modulus, and strain energy density of experimental groups. Data are mean ± sd. Statistical analyses were performed using unpaired two-tailed Student’s *t*-test (* *p* < 0.05). The X axis for (**b**,**c**) is silk fiber samples obtained from different diet additives and the control is the normal silk obtained from traditional diet without any additives. (Ser = serine, Tyr = tyrosine, SAA = sericin amino acid, FAA = Fibroin amino acid).

**Figure 5 materials-12-00014-f005:**
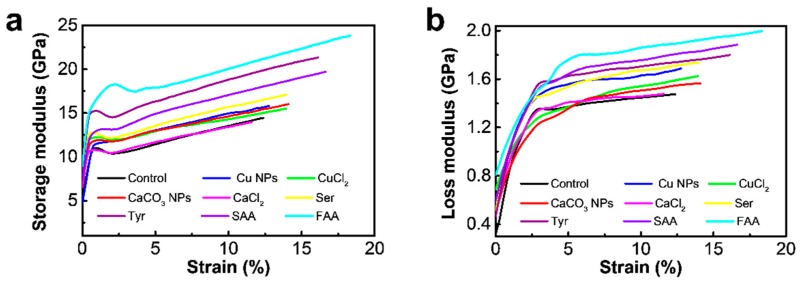
Continues dynamic analyses of silk fibers. (**a**) Typical storage modulus-strain curves. (**b**) Typical loss modulus-strain curves. (Ser = serine, Tyr = tyrosine, SAA = sericin amino acid, FAA = fibroin amino acid).

**Figure 6 materials-12-00014-f006:**
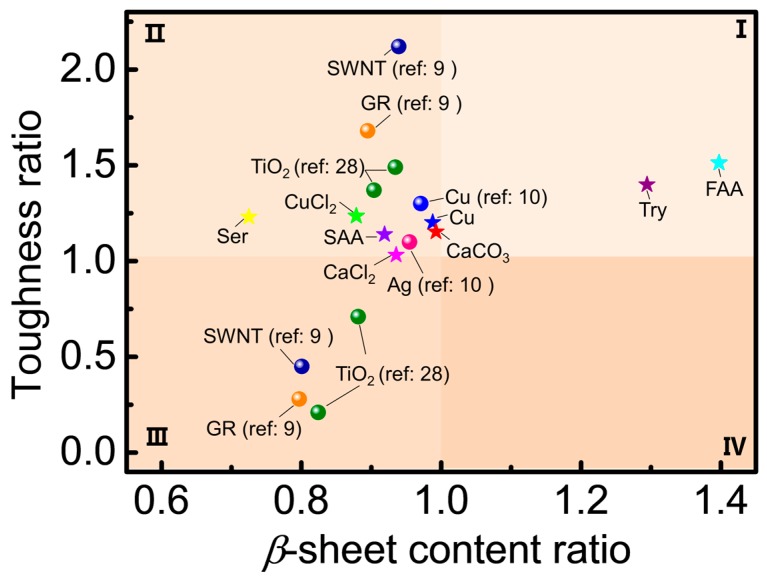
The relation of *β*-sheet content ratio and toughness ratio of silk fibers between modified and control group.

**Figure 7 materials-12-00014-f007:**
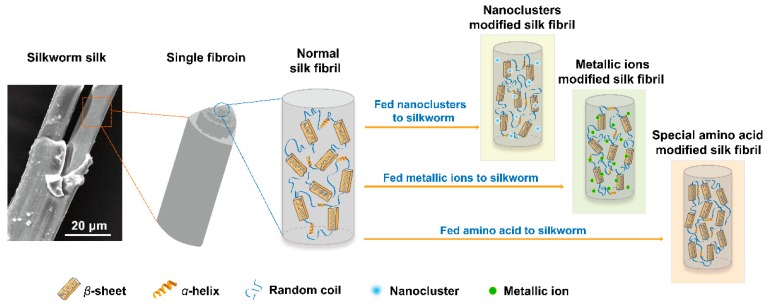
Schematic illustration of secondary structure of normal silk fibril and modified silk fibril in fibroin.

**Table 1 materials-12-00014-t001:** Characteristics of eight different additives.

Materials	Characteristics	Manufacturer
Nano Cu	Diameter: 20 nm, particles	Beijing DK Nano Technology Co., Ltd., Beijing, China
Nano CaCO_3_	Diameter: 20 nm, particles	
Copper(II) chloride (CuCl_2_)	Purity: ≥99.00%	Shanghai Aladdin Biochemical Technology Co., Ltd., Shanghai, China
Calcium chloride anhydrous (CaCl_2_)	Purity: ≥99.00%	
Serine (Ser)	L-Serine, food-grade	Huzhou Xintiansi Bio-tech Co., Ltd., Zhejiang province, China
Tyrosine (Tyr)	L-Tyrosine, food-grade	
Sericin amino acid (SAA)	Molecular weight: 500 Da	
Fibroin amino acid (FAA)	Molecular weight: 90 Da	

**Table 2 materials-12-00014-t002:** Typical degradation temperatures and residue masses of the silk fibers at 800 °C.

Samples	T_s_ (°C) ^a^	T_max_ (°C) ^b^	Residue at 800 °C (wt.%)
Control group	292.7	323.0	12.27
Cu NPs group	293.3	323.7	18.15
CuCl_2_ group	295.3	322.4	16.51
CaCO_3_ group	294.0	320.8	3.58
CaCl_2_ group	293.1	320.0	10.19
Serine (Ser) group	293.9	323.7	20.63
Tyrosine (Tyr) group	292.1	320.3	3.18
Sericin amino acid (SAA) group	297.1	323.8	18.65
Fibroin amino acid (FAA) group	290.8	320.8	15.14

^a^ T_s_ defined as the temperature at which the fibers start losing weight. ^b^ T_max_ defined as the temperature at which a maximum weight loss rate occurred.

**Table 3 materials-12-00014-t003:** Continues dynamic analysis of silk fibers. Data are mean ± sd. Statistical analyses were performed using unpaired two-tailed Student’s *t*-test (* *p* < 0.05).

Samples	Dynamic Strength (MPa)	Dynamic Modulus (GPa)	Storage Modulus (GPa)	Loss Modulus (GPa)
Control group	283 ± 31	10.6 ± 2.4	15.2 ± 2.2	1.32 ± 0.22
Cu NPs group	346 ± 50 *	13.5 ± 1.8 *	18.0 ± 2.2 *	1.69 ± 0.35 *
CuCl2 group	299 ± 61	11.8 ± 3.2	15.4 ± 2.1	1.48 ± 0.25
CaCO3 group	333 ± 57	12.7 ± 1.6	15.8 ± 2.1	1.45 ± 0.23
CaCl2 group	280 ± 46	11.3 ± 1.8	15.1 ± 2.6	1.42 ± 0.17
Serine (Ser) group	377 ± 49 *	13.1 ± 1.4 *	18.6 ± 2.1 *	1.69 ± 0.27 *
Tyrosine (Tyr) group	400 ± 41 *	13.5 ± 1.6 *	20.1 ± 2.2 *	1.69 ± 0.21 *
Sericin amino acid (SAA) group	386 ± 56 *	13.3 ± 1.7 *	19.4 ± 2.3 *	1.77 ± 0.24 *
Fibroin amino acid (FAA) group	459 ± 52 *	15.7 ± 1.9 *	23.3 ± 3.0 *	1.96 ± 0.27 *

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
