# Peer review of "Effect of Different Additives in Diets on Secondary Structure, Thermal and Mechanical Properties of Silkworm Silk"

_materials, 2018, doi:10.3390/ma12010014_

Round 1
Reviewer 1 Report
This manuscript reports how different additives in the silkworms’ diet may affect the mechanical and thermal stability properties of as-spun silk fibers. Overall, the text is well organized, the approach and the methods are well described. There are only few minor issues that should be addressed:
1) It would be useful to explain clearly in the introduction the motivation which moved the authors to select those specific eight additives in their study.
2) I would suggest the authors to include in their references fundamental studies about the mechanical properties of silkworm silk, such as Perez-Rigueiro et al 1998 J. Appl Polym Sci 70: 2439; Perez-Rigueiro et al 2000, J Appl Polym Sci 75: 1270; and recent studies about other strategies to improve the mechanical properties of silk fibers (with no chemical modification), see Berardo et al. 2016 Interface Focus 6: 20150060; Pantano et al 2016 Sci. Rep. 6: 18222; Cheng et al 2018 Polymers 10(11): 1214.
3) Within the amino acid materials, the silk fibers corresponding to Tyr-additive shows a significantly different behavior (the residue mass after 800°C is very small). Is it possible to explain such peculiar behavior?
4) English revision is suggested to correct few errors.
Author Response
Reviewer 1: Comments and Suggestions for Authors
This manuscript reports how different additives in the silkworms’ diet may affect the mechanical and thermal stability properties of as-spun silk fibers. Overall, the text is well organized, the approach and the methods are well described. There are only few minor issues that should be addressed:
Q (1): It would be useful to explain clearly in the introduction the motivation which moved the authors to select those specific eight additives in their study.
Answer: Thank you for your suggestion. We have added the reason for the selection of eight additives in the “Introduction” section in the revised manuscript.
Q (2): I would suggest the authors to include in their references fundamental studies about the mechanical properties of silkworm silk, such as Perez-Rigueiro et al 1998 J. Appl Polym Sci 70: 2439; Perez-Rigueiro et al 2000, J Appl Polym Sci 75: 1270; and recent studies about other strategies to improve the mechanical properties of silk fibers (with no chemical modification), see Berardo et al. 2016 Interface Focus 6: 20150060; Pantano et al 2016 Sci. Rep. 6: 18222; Cheng et al 2018 Polymers 10(11): 1214.
Answer: Thank you very much. We have added the related references about the mechanical properties and modifications of silkworm silk in the revised manuscript.
Q (3): Within the amino acid materials, the silk fibers corresponding to Tyr-additive shows a significantly different behavior (the residue mass after 800°C is very small). Is it possible to explain such peculiar behavior?
Answer: Thank you very much for your helpful suggestion. We have made the corresponding modifications to explain the lower residue mass of silk fibers obtained from feeding CaCO3 NPs-additive and Tyr-additive in the revised manuscript. The detailed and systematical studies of the increase or decrease of the residue mass of different modified silk fibers will be carried out in our further works.
Q (4): English revision is suggested to correct few errors.
Answer: Thank you very much for your helpful comments. To make the manuscript more readable, we have carefully checked the English of the manuscript and corrected the errors in the revised manuscript.
Reviewer 2 Report
In this article, the authors studied different substances used as silkworm diet. In particular, they analyzed the effect of these additives on thermal and mechanical properties of derived fibroin fibers. The paper could be of interest for the readers of Material but is necessary a revision to better explain the results.
I also suggest a revision of English. I recommended acceptance the article, after some modifications and integrations.
Comments:
- Introduction: why did you compared Cu-NPs and CuCl2? Why did you compared CaCO3-NPs and CaCl2? Free copper and calcium could have different effect on fibroin properties?
- Section 2.1: I suggest to the authors to add a table comprising all additives considered for the study. A summary table could improve the understanding of the work. Which type of amino acids were contained into the SAA and FAA?
- Figure 1: add the information about the X axis (e.g. diet additives) and the control (e.g. traditional diet without additives). To investigate the existence of homogeneous additive groups with similar effect on fibroin thermal and mechanical characteristics, could be interesting the use of Principal Component Analysis (PCA).
- Figure 7: it is not clear if the figure was related to the results obtained in this work. This schematic illustration could be used as graphical abstract or as representation of the experimental design. Please consider to shift the figure in the materials and methods section.
- Conclusion: this section could be modified indicating the best promising additive. It is not clear which type of compound better improve the quality of fibroin fibers.
Author Response
Reviewer 2: Comments and Suggestions for Authors
In this article, the authors studied different substances used as silkworm diet. In particular, they analyzed the effect of these additives on thermal and mechanical properties of derived fibroin fibers. The paper could be of interest for the readers of Material but is necessary a revision to better explain the results.
Q (1): I also suggest a revision of English. I recommended acceptance the article, after some modifications and integrations.
Answer: Thank you very much for your helpful suggestion. To make the manuscript more readable, we have carefully checked the English of the manuscript and made the corresponding modifications in the revised manuscript.
Q (2): Introduction: why did you compared Cu-NPs and CuCl2? Why did you compared CaCO3-NPs and CaCl2? Free copper and calcium could have different effect on fibroin properties?
Answer: Thank you very much for your helpful suggestion. We have added the reason for the selection of different additives in the “Introduction” part in the revised manuscript.
Q (3): Section 2.1: I suggest to the authors to add a table comprising all additives considered for the study. A summary table could improve the understanding of the work. Which type of amino acids were contained into the SAA and FAA?
Answer: Thank you very much. We have added a table in the revised manuscript to summary the detailed information of the eight kinds of additives. SAA and FAA are multi-amino acid directly obtained from the sericin and fibroin, respectively, of silk fibers.
Q (4): Figure 1: add the information about the X axis (e.g. diet additives) and the control (e.g. traditional diet without additives). To investigate the existence of homogeneous additive groups with similar effect on fibroin thermal and mechanical characteristics, could be interesting the use of Principal Component Analysis (PCA).
Answer: Thank you very much for your helpful suggestion. We have added the information about the X axis and the explanation of control group for all the figures in the revised manuscript.
Based on your helpful suggestion and considering the scope of this manuscript, the systematical studies on the homogeneous existence of the properties of silk fiber samples would be conducted in our future work.
Q (5): Figure 7: it is not clear if the figure was related to the results obtained in this work. This schematic illustration could be used as graphical abstract or as representation of the experimental design. Please consider to shift the figure in the materials and methods section.
Answer: Thank you very much for your useful suggestions. Figure 7 has been used for graphical abstract. In fact, Figure 7 not only shows the experimental design process but also illustrates the effect of different additives on protein conformational transitions in silk fibroin, which can be concluded from the numbers of the β-sheet and the existence form of additives in each silk fibril. Considering the function of Figure 7 and the framework of this study, we put this figure in the “Discussion” part to explain the modification mechanisms of different additives.
Q (6): Conclusion: this section could be modified indicating the best promising additive. It is not clear which type of compound better improve the quality of fibroin fibers.
Answer: Thank you very much for your helpful comments. We have modified the conclusion according to your suggestions.
Reviewer 3 Report
If the authors will make some reference about physical effects to fiber
formation,
this paper could be published after major revision.
Physiological point of view is necessary in this paper.
Basically CuCl2, nano Cu, and other metal ionic substrates are poisonous
for insects larvae. Even water has bad influence to larvae growth. There
is a possibility that such a poison affected the change of the
properties of silk.
Excessive amount of potassium ion in fibers which were from larvae whose
food contents additional Ser, Tyr, SAA ,and FAA implies the
physiological influence on this experimental results.
The point of this research is correlation between addition and content
of each metallic ion. If so, the authors should investigate a
relationship between the amount of additives and the content of
corresponding ion in fiber. I strongly recommend them to make additional
experiments.
About material.
FAA is listed as one of the eight materials, but there is no explanation in detail.
There are many amino acid in silk fibroin. Therefore more information is necessary.
Especially too much amino acid.
Author Response
Reviewer 3: Comments and Suggestions for Authors
Q (1): If the authors will make some reference about physical effects to fiber formation, this paper could be published after major revision.
Answer: Thank you very much for your useful comments. There are some references about physical effects to fiber formation such as the shear force and spinning speed mentioned in the Introduction section.
Q (2): Physiological point of view is necessary in this paper.
Answer: Thank you very much. Physiological point of view is very necessary to investigate the effects of additives on biological process of silkworms, however, it is beyond the scope of the present study and we will do further studies on it from the physiological point of view.
Q (3): Basically CuCl2, nano Cu, and other metal ionic substrates are poisonous for insects larvae. Even water has bad influence to larvae growth. There is a possibility that such a poison affected the change of the properties of silk.
Answer: Thank you very much for your useful comments. Yes, the toxicity of the additives has significant on silkworm growth and quality of silkworm cocoon and fibers, shown in our previous study [1]. Even through, the silkworms can produce enhanced silk fibers with good thermal and mechanical properties.
Q (4): Excessive amount of potassium ion in fibers which were from larvae whose food contents additional Ser, Tyr, SAA ,and FAA implies the physiological influence on this experimental results.
Answer: Thank you very much for your useful suggestions. The additional amino acid substances lead to a relative high potassium content means complex influences on silkworm’s biological or physiological process, such as the metabolism, metal ion transportation as well as the protein synthesis. It will be further systemically studied from biological views.
Q (5): The point of this research is correlation between addition and content of each metallic ion. If so, the authors should investigate a relationship between the amount of additives and the content of corresponding ion in fiber. I strongly recommend them to make additional experiments.
Answer: Thank you very much for your useful suggestions. It is very important to investigate the relationship between the amount of additives and the content of corresponding ion in silk fibers. We also did the corresponding research in our previous studies [1]. The detailed investigations about the relationship between the amount of these eight kinds of additives and the content of corresponding ion in fiber will be carried out in our further studies.
Q (6): About material. FAA is listed as one of the eight materials, but there is no explanation in detail. There are many amino acid in silk fibroin. Therefore more information is necessary. Especially too much amino acid.
Answer: Thank you very much for your useful suggestions. The detailed characteristics of FAA are listed in Table 1 in the revised manuscript. The FAA used in this study is a kind of fibroin protein with multi-amino acids directly obtained from degummed silk fibers.
Reference
[1] Cheng L., Huang H., Chen S., Wang W., Dai F., Zhao H. Characterization of silkworm larvae growth and properties of silk fibres after direct feeding of copper or silver nanoparticles. Mater Design, 2017, 129, 125-134.
Reviewer 4 Report
General Comments:
The nature of the discovery by the authors is amazing and they obtained promising results, both in their previous works and in this research. However, the results and discussion represented in this work need some general and major revisions.
The whole concept of this manuscript is similar to a report while the researchers tried to hypothesize those reported results (mainly in the discussion part) by referring to other non-relevant references. I highly recommend that the respectful researchers mainly focus on their current results (which are physicochemical properties of the final fibers) and justify those results in a scientific way. If they intend to obtain deeper understanding of the mechanisms in the macromolecular levels, they need to use more analytical techniques. In their discussion parts, they tried to justify their results in the macromolecular level using several non-relevant references, while they did not do any relevant analysis in the macromolecular level to make them capable of such assessments.
I divide my comments into 2 parts, the major comments and detailed comments:
Major Comments:
1) In the experimental part, the researchers used the reeling process to collect silk fibers and then they degummed those fibers. Did they use the degummed fibers for all the further techniques? Did they use a bundle of fibers for their analysis? They have to explain how they selected those fibers for mechanical analysis.
Also they mentioned the degumming process for 45 min and then washed the fibers with water. Why they used degumming process just one time, for 45 min? Are they trying to remove sericin in order to obtain Silk Fibroin? If so, it is possible that there still remained some sericin, as their degumming process might not be complete, and the remained sericin could influence their final results.
From my experimental point of view, and also articles such as “Effect of degumming condition on the solution properties and electrospinnablity of regenerated silk solution”, the degumming duration and the degumming agent can influence on the residual sericin, in a way that within 1 hours of degumming with Sodium Carbonate, the residual sericin will be around 8.2%. Moreover, it has been reported in articles such as “The impact of degumming conditions on the properties of silk films for biomedical applications” that sericin influences on the mechanical properties of degummed fibers. Hence, I am suspected that the resulting fibers in this study might be partially degummed silk fibers. Such results should be carefully interpreted.
2) The respectful researcher many times mentioned “proteins and conformational changes of proteins” within the whole text. Their results were mainly obtained from the characterization analysis of the degummed silk fibers which may contain both fibroin and sericin (as explained in previous issue). All the assessments including FTIR in their study is just some techniques to characterize the final obtained fibers and to confirm the influence of feeding additives on increasing or decreasing of the main substances of the silk fibers and the enhancement of final fiber properties. As they did not characterize any silk protein solutions in the macromolecular level, it is not enough to interpret the mechanism of feeding additive on the conformational changes of silk fibers. Nonetheless, they could not use these results to interpret or hypothesize the possible mechanisms involved in-vivo to obtain such final fibers.
If they intend to talk about the protein conformational changes in their final samples, they have to initially do the degumming process in the correct way (to get rid of sericin and obtain “Degummed Silk Fibroin Fibers”). Then dissolve their fibers in the relevant solvent for silk fibroin and characterize the final solutions in order to determine protein conformational changes. Many of the references they used to justify their interpretations, have used the regenerated fibroin solutions to detect the protein conformational changes.
3) Most of the hypothesis in the discussion part of this research are interpreted based on previous studies on silk fibroin. Nonetheless, I believe that the results presented by the respectful researchers might be influenced by silk sericin as well. So, I suggest that the researchers choose either of the following two ways:
- To keep their results as what they are, but change the whole context focusing on the final properties of degummed silk fibers (which might contain both fibroin and sericin) and discuss those results in more details. In this case they should avoid interpreting their results based on the references which worked just with silk fibroin (either in its solid form or solution form). Regarding their FTIR results, it might be influenced by both sericin and fibroin, in that case, the crystallinity and all the probable conformational changes could be used for the degummed silk fibers.
- Redo the degumming procedure, based on one of the relevant articles (2 or 3 times boiling in sodium carbonate for 30 min and washed with distilled water at the intervals). In this way, they might obtain the degummed silk fibroin fibers and do further characterization techniques on these fibers. In such case, the mechanical properties could be attributed to the silk fibroin fibers and is not influenced by any residual sericin component. They also can solve those silk fibroin fibers in the relevant solutions So, they can interpret those results in the macromolecular level with the relevant references. Still, they should be very careful with those references, as some of them characterized the regenerated fibroin or silk fibroin in solution.
Detailed comments:
Title:
I believe to change the title to the influence of additives on the mechanical and thermal properties of in-situ silkworm fibers, or something similar, and not referring to the conformational changes of proteins in the title.
Abstract:
Lines 19-20, better to re-write it.
Materials:
Did you obtain silk fibers from reeling process of silkworm cocoons? Did you analyze the experiments on these fibers or on the degummed ones?
ICP-MS measurements: Are the silk fibers used in this section similar to those degummed fibers you obtained in the materials section?
FTIR: In their FTIR results, it is better they bring the FTIR peak in wider area (around 1000 to 1800 cm-1) and then focus on the area between 1600 to 1720. Also, it might be better to bring the relevant references for the part of “straight baseline at the bottom of the peak”.
Results:
Page 4: The results in this section are mostly repeating the information presented in the figures, while it might be better to discuss those results in their relevant parts instead of mentioning them separately, in the discussion part. I think it is better to transfer some of the discussion parts to the relevant result part, eliminating any interpretation regarding the protein conformational changes in macromolecular level.
Page4, line146: “Particularly for Ca+2”, please re-write it, otherwise, it is very vague. Please check all the texts and change those parts mentioning the name of the additives, without mentioning the term of “fed additive” or “fibers modified by ..”. For example, in the aforementioned line, you have to write “particularly for the modified fibers by Ca+2” additive.
Page4, line 153: again the same as the previous one, “modified silk fibers by Ser… fed additives”.
Page5: I think the FTIR results of the already published manuscript by this group (Reference 8) are more convincible than what they presented in the relevant part of their current research. I recommend to rearrange this part based on their previous article. Also, it might be better if they bring the spectra from 1000 to 1800 and discuss all the amide (I, II and III) bands.
Page5, line 191-193: it is better to re-write the figure caption, specifically “(b)” which represents the content % after deconvolution, and (d) which is the crystallinity percentage of silk fibers.
Page 6: All this page mostly contain reports of the information represented in Figure 3, without discussing those results in more details. It might be better to discuss these results considering the FTIR and XRD results in the previous sections and justify the thermal behavior with the crystallinity content of the fibers. I think it might be better to present Figure 3c and 3d in a table.
Page6, line 197: “thermal degradation weight loss”, what is that? Is it the temperature in which the fibers start to losing weight? If so, better to re-write it to a more understandable phrase.
Page6, line 207: better to mention “modified silk fibers by fed additive SER and SAA”.
Page6: Figure 3b, for some of the samples, it shows a peak between 600-800 degree. Better to discuss the reason.
Pages 6 and 7: again you have just reported the information in Figure 4, without discussing those results in more details. Then in page 8 lines 234 – 244 you explained some obvious explanation for the terms used in Figure 4. However, there is no need to explain those terms and you have to discuss those terms with your data in Figure 4.
Page 8: you can discuss the results of Figure 5 with your XRD, FTIR and TGA results. All these results are related to each other. These are the final characteristics of the fibers and you can justify them with each other. Again in this part you report all the information in Figure5 without discussing the data in more details.
Page 9: better to bring Figure 5c and 5d in a table.
Page 9: most of the discussion part is unacceptable, since the researchers tried to discuss the macromolecular levels, while they just did some physicochemical properties on the final fibers. At lease they have to do some analysis in the macromolecular level to be able to interpret their reported results with those deeper analyses. Otherwise, their hypothesis, specifically what they presented in Figure 7, is unacceptable.
Page9, line 279-281: such interpretation is unacceptable, because the research to which they referred (reference 43), was a research being done on regenerated silk fibroin and in the solution.
Page9, line283-284: the same as previous comment, it is not possible to interpret these results for the macromolecular level analysis of “inhibiting beta sheet transition in-vivo”.
Page9, line 292: again they tried to justify their results of NPs to hinder the beta sheet formation, through referring to some references, which is not proper way for a scientific article.
Page 10, line 316-329: This part might be a good part for the discussion which could be improved and extended, bringing all the results of physicochemical properties of the fibers. Better to mainly focus on the final properties of the fibers in a scientific way, while avoiding to express some non-relevant references to justify the macromolecular level from which there is no obtained information.
All of the remained parts of their result is just discussing the topics from which they do not have any analysis in hand. Their schematic figure 7 is based on their hypothesis in the macromolecular level, from non-relevant references, which is not necessary.
All in all, I again emphasize that you have to just focus on your gained results, discuss them scientifically, relate them together and compare those properties of obtained fibers with each other and with the controlled sample.
Author Response
Reviewer 4: Comments and Suggestions for Authors
The nature of the discovery by the authors is amazing and they obtained promising results, both in their previous works and in this research. However, the results and discussion represented in this work need some general and major revisions.
Major Comments:
Q (1): In the experimental part, the researchers used the reeling process to collect silk fibers and then they degummed those fibers. Did they use the degummed fibers for all the further techniques? Did they use a bundle of fibers for their analysis? They have to explain how they selected those fibers for mechanical analysis.
Answer: Thank you very much for your useful suggestions. We used the degummed silk fibers for all the further experiments and the fibers used for mechanical properties are single fibers. We select the single fibers on a fabric blackboard. Before the tensile test, the average diameter of the fibers was measured using a laser scanning microscope. So, the bundle silk fibers can be avoided in this approach.
Q (2): Also they mentioned the degumming process for 45 min and then washed the fibers with water. Why they used degumming process just one time, for 45 min? Are they trying to remove sericin in order to obtain Silk Fibroin? If so, it is possible that there still remained some sericin, as their degumming process might not be complete, and the remained sericin could influence their final results.
From my experimental point of view, and also articles such as “Effect of degumming condition on the solution properties and electrospinnablity of regenerated silk solution”, the degumming duration and the degumming agent can influence on the residual sericin, in a way that within 1 hours of degumming with Sodium Carbonate, the residual sericin will be around 8.2%. Moreover, it has been reported in articles such as “The impact of degumming conditions on the properties of silk films for biomedical applications” that sericin influences on the mechanical properties of degummed fibers. Hence, I am suspected that the resulting fibers in this study might be partially degummed silk fibers. Such results should be carefully interpreted.
Answer: Thank you very much for your useful suggestions. I am sorry for the mistake in the experimental section and we have modified it in the revised manuscript. In fact, the degummed silk fibers were obtained by degumming twice in 0.5% Na2CO3 solution for 45 min then washed several times. The degummed silk fibers are also carefully checked whether the degumming process are complete before further experiments.
Q (3): The respectful researcher many times mentioned “proteins and conformational changes of proteins” within the whole text. Their results were mainly obtained from the characterization analysis of the degummed silk fibers which may contain both fibroin and sericin (as explained in previous issue). All the assessments including FTIR in their study is just some techniques to characterize the final obtained fibers and to confirm the influence of feeding additives on increasing or decreasing of the main substances of the silk fibers and the enhancement of final fiber properties. As they did not characterize any silk protein solutions in the macromolecular level, it is not enough to interpret the mechanism of feeding additive on the conformational changes of silk fibers. Nonetheless, they could not use these results to interpret or hypothesize the possible mechanisms involved in-vivo to obtain such final fibers.
If they intend to talk about the protein conformational changes in their final samples, they have to initially do the degumming process in the correct way (to get rid of sericin and obtain “Degummed Silk Fibroin Fibers”). Then dissolve their fibers in the relevant solvent for silk fibroin and characterize the final solutions in order to determine protein conformational changes. Many of the references they used to justify their interpretations, have used the regenerated fibroin solutions to detect the protein conformational changes.
Answer: Thank you very much for your useful suggestions. The materials used in this study are fibroin fiber without any sericin. The conformational changes of proteins or the conformational transitions mentioned in this manuscript are the indirect reflection of the secondary structures of silk fibers.
Q (4): Most of the hypothesis in the discussion part of this research are interpreted based on previous studies on silk fibroin. Nonetheless, I believe that the results presented by the respectful researchers might be influenced by silk sericin as well. So, I suggest that the researchers choose either of the following two ways:
To keep their results as what they are, but change the whole context focusing on the final properties of degummed silk fibers (which might contain both fibroin and sericin) and discuss those results in more details. In this case they should avoid interpreting their results based on the references which worked just with silk fibroin (either in its solid form or solution form). Regarding their FTIR results, it might be influenced by both sericin and fibroin, in that case, the crystallinity and all the probable conformational changes could be used for the degummed silk fibers.
Redo the degumming procedure, based on one of the relevant articles (2 or 3 times boiling in sodium carbonate for 30 min and washed with distilled water at the intervals). In this way, they might obtain the degummed silk fibroin fibers and do further characterization techniques on these fibers. In such case, the mechanical properties could be attributed to the silk fibroin fibers and is not influenced by any residual sericin component. They also can solve those silk fibroin fibers in the relevant solutions So, they can interpret those results in the macromolecular level with the relevant references. Still, they should be very careful with those references, as some of them characterized the regenerated fibroin or silk fibroin in solution.
Answer: Thank you very much for your useful suggestions. The materials used in this study are totally degummed silk fibers or the silk fibroin. The sercin were removed completely and the experimental results are not influenced by the residual sericin component.
Detailed comments:
Q (5): Title: I believe to change the title to the influence of additives on the mechanical and thermal properties of in-situ silkworm fibers, or something similar, and not referring to the conformational changes of proteins in the title.
Answer: Thank you very much for your useful suggestions. We have changed title into “Effect of Different Additives in Diets on Secondary Structure, Thermal and Mechanical Properties of Silkworm Silk” in the revised manuscript.
Q (6): Abstract: Lines 19-20, better to re-write it.
Answer: Thank you very much for your useful suggestions. We have rewrite the sentences in the abstract.
Q (7): Materials: Did you obtain silk fibers from reeling process of silkworm cocoons? Did you analyze the experiments on these fibers or on the degummed ones?
Answer: Thank you very much for your useful suggestions. We obtained the silk fiber from reeling process of silkworm cocoons. We also chose silk fibers from five cocoons with almost same lengths and weights in the nearly same position (from the 200th to 600th meter) as testing samples to reduce the variability at the intraspecific and intra individual levels. Then, these fibers were used for further analysis after degumming.
Q (8): ICP-MS measurements: Are the silk fibers used in this section similar to those degummed fibers you obtained in the materials section?
Answer: Thank you very much. All the fibers used in this section and other experiments are the degummed fiber samples.
Q (9): FTIR: In their FTIR results, it is better they bring the FTIR peak in wider area (around 1000 to 1800 cm-1) and then focus on the area between 1600 to 1720. Also, it might be better to bring the relevant references for the part of “straight baseline at the bottom of the peak”.
Answer: Thank you very much. The FTIR spectra of silk fibers around 1000 to 1800 cm-1 are shown in the revised manuscript and the typical FTIR peak are almost in the same position. So, we mainly focus on the quantitative analysis in amide I for the content of β-sheet, random coil and β-turn according to the relevant references.
Results:
Q (10): Page 4: The results in this section are mostly repeating the information presented in the figures, while it might be better to discuss those results in their relevant parts instead of mentioning them separately, in the discussion part. I think it is better to transfer some of the discussion parts to the relevant result part, eliminating any interpretation regarding the protein conformational changes in macromolecular level.
Answer: Thank you very much. We have transferred the discussion parts to the relevant result parts and did some revisions about the discussions.
Q (11): Page4, line146: “Particularly for Ca2+”, please re-write it, otherwise, it is very vague. Please check all the texts and change those parts mentioning the name of the additives, without mentioning the term of “fed additive” or “fibers modified by ..”. For example, in the aforementioned line, you have to write “particularly for the modified fibers by Ca2+” additive.
Answer: Thank you very much for your help suggestions. We have modified the corresponding confused words in the revised manuscript.
Q (12): Page4, line 153: again the same as the previous one, “modified silk fibers by Ser… fed additives”.
Answer: Thank you very much for your help suggestions. We have modified the sentence in the revised manuscript.
Q (13): Page5: I think the FTIR results of the already published manuscript by this group (Reference 8) are more convincible than what they presented in the relevant part of their current research. I recommend to rearrange this part based on their previous article. Also, it might be better if they bring the spectra from 1000 to 1800 and discuss all the amide (I, II and III) bands.
Answer: Thank you very much. We have changed the FTIR spectra in Figure 2 in the revised manuscript.
Q (14): Page5, line 191-193: it is better to re-write the figure caption, specifically “(b)” which represents the content % after deconvolution, and (d) which is the crystallinity percentage of silk fibers.
Answer: Thank you very much. We have rewrite the figure caption in Figure 2 in the revised manuscript.
Q (15): Page 6: All this page mostly contain reports of the information represented in Figure 3, without discussing those results in more details. It might be better to discuss these results considering the FTIR and XRD results in the previous sections and justify the thermal behavior with the crystallinity content of the fibers. I think it might be better to present Figure 3c and 3d in a table.
Answer: Thank you very much for your help suggestions. We have changed figure 3c and 3d into table.
Q (16): Page6, line 197: “thermal degradation weight loss”, what is that? Is it the temperature in which the fibers start to losing weight? If so, better to re-write it to a more understandable phrase.
Answer: Thank you very much. We have rewrite the phrase “thermal degradation weight loss” in the revised manuscript.
Q (17): Page6, line 207: better to mention “modified silk fibers by fed additive SER and SAA”.
Answer: Thank you very much for your help suggestions. We have modified the sentence in the revised manuscript.
Q (18): Page6: Figure 3b, for some of the samples, it shows a peak between 600-800 degree. Better to discuss the reason.
Answer: Thank you very much. We have added some discussions about the peaks between 600-800 degree
Q (19): Pages 6 and 7: again you have just reported the information in Figure 4, without discussing those results in more details. Then in page 8 lines 234-244 you explained some obvious explanation for the terms used in Figure 4. However, there is no need to explain those terms and you have to discuss those terms with your data in Figure 4.
Answer: Thank you very much. We have added some discussions about the mechanical properties for different modified silk fibers.
Q (20): Page 8: you can discuss the results of Figure 5 with your XRD, FTIR and TGA results. All these results are related to each other. These are the final characteristics of the fibers and you can justify them with each other. Again in this part you report all the information in Figure5 without discussing the data in more details.
Answer: Thank you very much for your help suggestions. We have added some detailed discussions in this part in the revised manuscript.
Q (21): Page 9: better to bring Figure 5c and 5d in a table.
Answer: Thank you very much for your help suggestions. We have changed figure 5c and 5d into table.
Q (22): Page 9: most of the discussion part is unacceptable, since the researchers tried to discuss the macromolecular levels, while they just did some physicochemical properties on the final fibers. At lease they have to do some analysis in the macromolecular level to be able to interpret their reported results with those deeper analyses. Otherwise, their hypothesis, specifically what they presented in Figure 7, is unacceptable.
All of the remained parts of their result is just discussing the topics from which they do not have any analysis in hand. Their schematic figure 7 is based on their hypothesis in the macromolecular level, from non-relevant references, which is not necessary.
Answer: Thank you very much for your help suggestions. Figure 7 is a schematic illustration of the secondary structure of silk fibril. We try to use the changes of β-sheet content in the silk fibril to reveal the effects of additive types on the secondary structure of silk fiber, then relate to the thermal and mechanical properties.
Q (23): Page9, line 279-281: such interpretation is unacceptable, because the research to which they referred (reference 43), was a research being done on regenerated silk fibroin and in the solution.
Answer: Thank you very much for your help suggestions. Both of the original silk fibroin and regenerated silk fibroin are proteins. Most of the researches were carried out to investigate the spinning process and formation of silks by using regenerated silk fibroin. It is a mechanism study for the silk fibers. So, the research results from the regenerated silk fibroin would be a references for the silk fibers.
Q (24): Page9, line283-284: the same as previous comment, it is not possible to interpret these results for the macromolecular level analysis of “inhibiting beta sheet transition in-vivo”. Page9, line 292: again they tried to justify their results of NPs to hinder the beta sheet formation, through referring to some references, which is not proper way for a scientific article.
Answer: Thank you very much for your help suggestions. We made a conclusion of the “inhibiting beta sheet transition in-vivo” from the decrease of the β-sheet content and crystallinity, and it is also mentioned in some references.
Q (25): Page 10, line 316-329: This part might be a good part for the discussion which could be improved and extended, bringing all the results of physicochemical properties of the fibers. Better to mainly focus on the final properties of the fibers in a scientific way, while avoiding to express some non-relevant references to justify the macromolecular level from which there is no obtained information.
All in all, I again emphasize that you have to just focus on your gained results, discuss them scientifically, relate them together and compare those properties of obtained fibers with each other and with the controlled sample.
Answer: Thank you very much for your help suggestions. We have modified all the discussions in a proper way in the revised manuscript according to your comments.
Round 2
Reviewer 4 Report
The manuscript improved regarding the content, however, still a huge English language editing is needed. Besides my comments in the following, I recommend the authors to take a more careful and deeper look throughout the text and revise its English language. Some of these comments are as the following:
- Line 51: Please eliminate " , " after "that" in the sentence "indicated that through disrupting .."
- Line 54: " in the spinning duct, and then lead to the .." better to change to " in the spinning duct, leading to the significant ...".
- Line 61: "on the conformational transition, and thermal and mechanical.. "
- Lines 86-87: I don't think it is necessary to bring that sentence. As long as you do the degumming procedure the same as other references, it is enough. It might bring questions to the readers about the methods you've used to check the complete degumming procedure. I think better not to mention it.
-Table 1 does not look proper in the version I've received. the texts are into each other and the table lines are not complete. Also, table should be completely transferred to the next page. Please check it again
- Lines 187-188: Please revise that sentence, it is not clear. do you mean "... the basic secondary structure of silk fibroin.."?
- Line 190: "deconvolutions of the amide I region.."
- Line 197: "to distinguish the contents of the random coil.."
- Line 211: "Compared with the control silk fiber with an overall crystallinity of .."
- Line 249: "which could be will be further studies systematically"
- Table 2: in the third column, please put the numbers with 2 digit after .
- Lines 349-350: ".. on secondary structure of silk fibers .. β-sheet content of silk fibril is used to present the secondary structure".
- Line 380: "are the best promising additives.... are not only cheap and safe enough.... both crystalinity and mechanical.."
- Line 381: "Besides, the effects mechanism effects of.."
Author Response
Response to Reviewer 4 Comments
Reviewer 4: Comments and Suggestions for Authors
The manuscript improved regarding the content, however, still a huge English language editing is needed. Besides my comments in the following, I recommend the authors to take a more careful and deeper look throughout the text and revise its English language. Some of these comments are as the following:
- Line 51: Please eliminate " , " after "that" in the sentence "indicated that through disrupting .."
- Line 54: " in the spinning duct, and then lead to the .." better to change to " in the spinning duct, leading to the significant ...".
- Line 61: "on the conformational transition, and thermal and mechanical.. "
- Lines 86-87: I don't think it is necessary to bring that sentence. As long as you do the degumming procedure the same as other references, it is enough. It might bring questions to the readers about the methods you've used to check the complete degumming procedure. I think better not to mention it.
-Table 1 does not look proper in the version I've received. the texts are into each other and the table lines are not complete. Also, table should be completely transferred to the next page. Please check it again
- Lines 187-188: Please revise that sentence, it is not clear. do you mean "... the basic secondary structure of silk fibroin.."?
- Line 190: "deconvolutions of the amide I region.."
- Line 197: "to distinguish the contents of the random coil.."
- Line 211: "Compared with the control silk fiber with an overall crystallinity of .."
- Line 249: "which could be will be further studies systematically"
- Table 2: in the third column, please put the numbers with 2 digit after .
- Lines 349-350: ".. on secondary structure of silk fibers .. β-sheet content of silk fibril is used to present the secondary structure".
- Line 380: "are the best promising additives.... are not only cheap and safe enough.... both crystalinity and mechanical.."
- Line 381: "Besides, the effects mechanism effects of.."
Answer: Thank you very much for your useful comments. We have carefully checked the whole manuscript and improved the English. The revised portions are all marked in green in the revised manuscript.